Methods

# Scalable workflows for high-throughput respirometry

Corey A Osto[1,2] ⓘ, Eugene V Mosharov[3,4], Ayelet M Rosenberg[3], Martin Picard[3,4,5,6], Linsey Stiles[1,2], Orian Shirihai[1,2] ⓘ

**Mitochondrial respirometry, the measurement of oxygen consumption rate (OCR) by the electron transport chain (ETC), is a cornerstone of mitochondrial biology and the gold standard for measurements of mitochondrial function. However, existing respirometry methodologies are poorly suited for large-scale studies and high-throughput applications, ultimately limiting the applicability of these methods. This limitation necessitates new methodologies, which are more easily scaled as mitochondrial studies become more complex and diverse. In this study, we detail a respirometry approach we have developed for high-throughput applications including optimized plate layouts, volume-based sample normalization, robust control selection, and automated data processing and quality control. Furthermore, we validate these methodologies across a respirometry study running 703 human brain samples, totaling more than 10,000 data points, which underwent our automated data processing and quality control techniques. Our workflow streamlines assay preparation, execution, and analysis to make respirometry scalable, while reducing operator burden and preserving data integrity. With this study, we provide a transferable blueprint for high-throughput respirometry as the mitochondrial biology field and the studies within it continue to expand in scale.**

## Introduction

Mitochondrial research has grown exponentially in the last two decades, and at the forefront of mitochondrial studies are measurements of mitochondrial function, specifically respirometry. Respirometry is the measurement of oxygen consumption rate (OCR) by the mitochondria, which occurs through Complex IV of the electron transport chain (ETC). This technique represents the gold standard of measurements of mitochondrial function for two primary reasons: (1) oxygen consumption is an output for which we have advanced ways of very accurately measuring small changes in

the partial pressure of oxygen in vitro, and (2) respirometry represents an integrated measurement of energy metabolism, reflecting changes in the energy-producing processes, which come before the ETC, and also the energy-consuming processes, which result from the tight coupling of the ETC with ATP production. With these considerations in mind, respirometry is a technique that has become a primary building block in the understanding of mitochondrial biology.

Despite being such an integral measurement to understanding whole tissue metabolism and bioenergetics, current respirometry methodologies have multiple limitations. Notable among these limitations is the necessity for tissue samples to be obtained and processed freshly, within hours of tissue isolation. This limits the scalability of respirometry and prevents larger clinical studies from biobanking material. Furthermore, rerunning the samples and/or transporting them to locations with the appropriate expertise and equipment to perform respirometry is problematic. With this in mind, we recently developed an approach that allows for the measurements of mitochondrial respiratory capacity in samples, which were previously frozen (Acin-Perez et al, 2020; Osto et al, 2020). In short, the respirometry in previously frozen samples (RIFS) assay adds back the principal components of ETC that were lost during freezing and thawing of the mitochondria (Acin-Perez et al, 2020; Osto et al, 2020). This technique overcomes the necessity for fresh tissue samples and allows for biobanking of large numbers of samples for respirometry measurements, overcoming one of the main factors limiting the scalability of respirometry (though the measurement of coupled respiration can no longer be measured in previously frozen samples). Until recently, RIFS has been employed on relatively small datasets, which is considered low-throughput compared with many "omics" and screening techniques used today (Rogers et al, 2011; Fuchs et al, 2023; Vekaria et al, 2024). Therefore, RIFS studies have relied on the low-throughput experimental planning, normalization, and analysis techniques, which are most commonly used for respirometry (Acin-Perez et al, 2020; Osto et al, 2020). To scale RIFS for the analysis of hundreds or even thousands of samples, new experimental considerations must be made regarding the planning, technical, and analytical changes required to achieve this type of large scalability.

[1]Department of Medicine, Endocrinology, University of California, Los Angeles, Los Angeles, CA, USA [2]Department of Molecular and Medical Pharmacology, David Geffen School of Medicine, University of California, Los Angeles, Los Angeles, CA, USA [3]Department of Psychiatry, Divisions of Molecular Therapeutics and Behavioral Medicine, Columbia University Irving Medical Center, New York, NY, USA [4]New York State Psychiatric Institute, New York, NY, USA [5]Department of Neurology, H. Houston Merritt Center, Columbia Translational Neuroscience Initiative, Columbia University Irving Medical Center, New York, NY, USA [6]Robert N. Butler Columbia Aging Center, Columbia University Mailman School of Public Health, New York, NY, USA

Correspondence: oshirihai@mednet.ucla.edu

Performing respirometry on such a large number of samples, especially when blinded to the sample types, requires novel experimental considerations. These include the following:

- Optimizing sample loading and plate layouts for efficient handling and data processing in high throughput.
- Determining the proper controls to account for batch variability.
- Altering sample loading normalization and techniques to be suited for large sample numbers and to minimize sample degradation.
- Developing an automated respirometry data handling and analysis protocol.

- Developing an automated and unbiased method for quality control of respirometry sample data.

In this study, we describe the techniques we developed for scaling respirometry and validate them using a novel dataset of more than 10,000 data points. We detail them with consideration to guiding future respirometry studies of this and larger scale.

## Technical Considerations for High-throughput Respirometry

**Technical challenges to consider for high-thoughput respirometry and their solutions.**

| Challenges to consider | Parameters to consider when choosing a solution | Solutions to consider for your study |
|---|---|---|
| How large is the scale of the experiment? | How many samples? Are the study team blinded to the sample/tissue type? | When possible, blinded sample are recommended using number codes in an ascending order. We recommend to pretest your workflow and samples to assess how your protocol will adapt to high-throughput techniques. |
| | How many technical replicates will be needed? | Each added replicate increases cost and labor when expanded to hundreds of samples. In scenarios where samples may spatially neighbor each other, or if pretesting data indicate that samples perform well across replicates, technical duplicates may be sufficient. We recommend duplicates be run on separate plates. |
| | What study controls can be used to assess the quality of the data? | We recommend aliquots of isolated mitochondria or tissue homogenates as technical controls for each study plate. We would also recommend a low-activity, medium-activity, and high-activity sample pool to better assess sample performance and plate variation to establishment of quality control metrics. |
| | Can a subset of the samples be used to test if your process is streamlined? | Test samples should be assayed to assess loading concentrations and workflow feasibility. |
| Are the samples going to be energetically similar? | How will sample loading concentration be normalized? | Pretest samples to determine volume or total protein for sample loading. When the samples are expected to have similar activities, determine that the samples are within the signal-to-noise range of the instrument. When samples in the study may be of different activity, determine a volume which places most of your samples within the signal-to-noise range of the instrument. These samples may require more reruns. |
| | Can the sample plates be loaded with a multichannel pipette? | Loading with a normalized volume allows for multichannel sample loading, making your workflow better for high-throughput applications. |
| | How will these choices effect your data at the end of the study? | Normalizing via a standard volume will often require later quality control metrics to identify samples to be rerun and correct for potential artifacts resulting from low/high oxygen consumption. |
| How will the samples be run? | How many instruments does the study team have access to? | Having multiple instruments can further streamline workflow by running plates in duplicate. Determine how many samples can be run in a day to optimize the study's workflow. |
| | How can freeze–thaw cycles and temperature fluctuations be minimized? | Avoid excessive freezing and thawing of your samples. Establish workflow that will minimize multiple freeze–thaw cycles keeping in mind that reruns may be necessary. Always keep samples at –80°C or, when preparing for RIFS, on ice. |
| | How will the data look when the study is finished? | Design a standard plate layout that will streamline analysis. Maintaining consistent plate loading across all runs allows for easier data handling when consolidating and analyzing the data. |

Seahorse experiments rely on multiple technical considerations to ensure that samples run consistently and within the signal-to-noise range of the instrument. This section details how to handle the scaling of the large number of blinded respirometry samples, the choice of control samples, plate layouts, and data analysis.

### Sample loading with respect to normalization of tissue content/mitochondrial activity

One of the most important factors to ensure successful and trustworthy respirometry data is to ensure that the samples operate within the signal-to-noise range of the Seahorse platform (Divakaruni & Martin, 2022). For isolated mitochondria and RIFS, this is typically achieved through the normalization of sampling loading, where total protein is measured for each sample and samples are individually loaded onto a Seahorse sample plate at different volumes to an equivalent total protein concentration. Although in low-throughput applications, it is feasible to use a single-channel pipette to separately load each sample individually, doing this for hundreds or even thousands of samples is less feasible. In addition, in sample sets, which are blinded to the operator and/or consist of multiple sample types (which may be energetically different), normalizing sampling loading via this traditional method is not practical. With these considerations in mind, respirometry for high-throughput applications and/or large datasets requires a more optimal method for sample loading. To overcome this technical hurdle, we show that loading using a single sample volume for all samples and normalizing to protein content after a respirometry run are viable for high-throughput applications.

Previous studies measuring mitochondrial activity in the brain have shown that gray matter and white matter in the brain differ greatly with regard to mitochondrial energetics (Tomasi et al, 2013; Fecher et al, 2019; He et al, 2022; Rosenberg et al, 2023). With this in mind, a collection of gray matter, white matter, and mixed brain matter (an intermediate of gray and white) represents a sample set, which would be difficult to run together in high-throughput respirometry applications. Fig 1A showcases the traditional method of normalization, where different protein concentrations produce outcomes within the signal-to-noise range of the instrument. Although this method produces reliable results, different protein concentrations are required for each sample type; therefore, in a high-throughput or blinded sample set, this method would not be viable. Picking a suitable amount for constant volume sample loading is ideally done during pretesting of the sample set, but certain considerations must be satisfied; namely, the most energetic samples (in this example, gray brain matter) must not exceed the upper limit of detection, and the least energetic samples (in this example, white brain matter) must not be below the lower limit of detection of the instrument.

In Fig 1B, we see the respirometry traces of gray matter control samples run at 10, 15, and 20 $\mu l$. At 20 $\mu l$ of sample, we begin to observe artifacts representative of a respirometry sample, which has been loaded at too high of a concentration. After TMPD/ascorbate injection, we observe an expected increase in OCR consistent with an increase in sample concentration, but at the second measurement, we see that this increased OCR is not sustained. Taking a closer look

at the oxygen levels within the wells of these samples (Fig 1C), one can see that at 20 $\mu l$ of sample, oxygen is depleted too rapidly, resulting in an artifact colloquially known as "J-curving" (discussed in Osto et al [2020], troubleshooting of the RIFS assay). When oxygen depletion occurs before the end of a measurement (rates over ~500 pmol $O_2$/min on the Seahorse XF96), the Seahorse instrument algorithm will no longer be able to properly measure the slope of the partial pressure of oxygen over time, resulting in incorrect and often inconsistent oxygen consumption measurements (Divakaruni & Martin, 2022). These data showcase the potential problems associated with loading respirometry samples at too high of a concentration and reveal that 20 $\mu l$ of tissue lysate is too much for this set of samples. In cases where pretesting cannot and/or is not performed, quality control steps may be taken to identify samples which may be exhibiting signs of oxygen depletion (discussed in the Discussion section of this manuscript).

In Fig 1D, we see the respirometry traces of white matter control samples run at 10, 15, and 20 $\mu l$. With these lower oxygen-consuming samples, we observe the opposite trend where at 15 and 20 $\mu l$ loading volumes, the traces were sensitive to mitochondrial substrates and inhibitors, but at 10 $\mu l$, we see unstable responses to injection compounds resulting in decreased reproducibility of technical replicas. Oxygen consumption, which is too low (below the limit of detection of the instrument), will appear as a downward slope of oxygen consumption over time, which is barely distinguishable from noise.

Based on these pretesting experiments, we would decide to run all samples at 15 $\mu l$. At this concentration, the white matter samples remained sensitive to all compounds and could be accurately detected, whereas the gray matter samples did not show any sign of consuming too much oxygen. While loading at a standard volume works for most of the samples, some factors should be considered before applying this technique. This approach is made much easier when working with samples of consistent protein concentration, a factor that should be considered when acquiring and processing samples for a study. Although this will still work with samples of less consistent protein content, it will likely require loading samples of similar protein concentration together or will require rerunning samples at different concentrations to ensure the data are within the signal-to-noise range of the instrument. This system only works in tandem with a quality control system that can identify samples with OCR values above or below the detection of the Seahorse assay so that they may be rerun. We will discuss how to deal with quality control of these types of samples in the Analytical Considerations section of this document.

### Control samples

For studies with large sample sets, selecting the proper control samples is critical not only to assess run performance, but also to later determine appropriate quality control criteria. At minimum, each Seahorse experiment should contain the following controls: background wells (filled with buffer but no sample) and a positive control, which has been proven to perform well in the chosen assay (an example for the RIFS respirometry platform would be mouse liver mitochondria [Osto et al, 2020]). Optimally, control samples (matching the sample type of the cohort), which are low, medium,

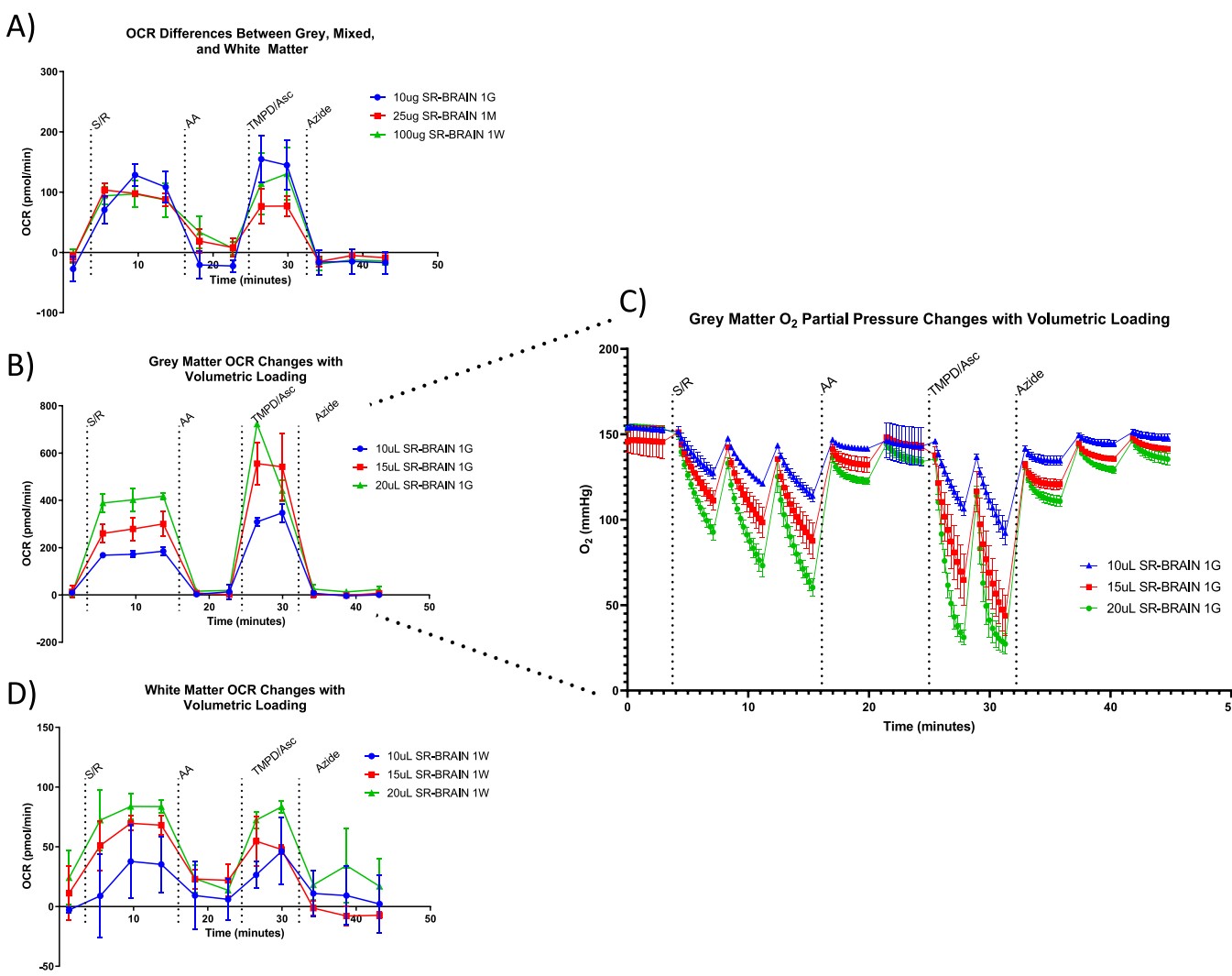

**Figure 1. Sample loading by volume as a technique for high-throughput respirometry using energetically different sample tissues.**
**(A)** Representative respirometry traces of brain gray matter (blue), white matter (green), and mixed matter (red) samples run using the traditional total protein loading method: gray matter loaded at 10 $\mu$g, mixed matter loaded at 25 $\mu$g, and white matter loaded at 100 $\mu$g show similar oxygen consumption rate. Though this method assures that all samples are run within the signal-to-noise range of the Seahorse instrument, it requires prior knowledge of the sample type and manual loading of the samples, two factors that are often not feasible in a blinded large-scale study. **(B)** Gray matter brain samples at differential sample loading volume (10, 15, and 20 $\mu$l). Although 10 and 15 $\mu$l loading has no artifacts, at 20 $\mu$l inconsistency in TMPD/Asc measurements was observed, indicative of sample overloading. **(B, C)** Oxygen partial pressure trace of the respirometry results shown in (B). At 20 $\mu$l, we observe "J-curving," an artifact of sample overloading where excessive oxygen consumption is preventing proper re-equilibration of oxygen levels within the Seahorse well. **(D)** White matter brain samples run at differential volume (10, 15, and 20 $\mu$l). At 10 $\mu$l loading, decreased sample reproducibility because of the low signal-to-noise range of the Seahorse instrument is observed. (G, gray brain matter; W, white brain matter; M, mixed brain matter). (SR, succinate/rotenone; the post-succinate/rotenone injection oxygen consumption rate is representative of a Complex II activity reading).

and high oxygen-consuming samples, should also be added. These controls ensure that the measurements are within the dynamic range of the Seahorse platform, serve as additional positive controls, and can be used to account for plate-to-plate variance during data analysis.

We found through our testing of these methodologies that for RIFS-based respirometry techniques, it is important that controls and sample material are aliquoted in advance such that they will have the same number of freeze–thaw cycles at the time they are run for the study. With larger numbers of freeze–thaw cycles, samples will exhibit lower oxygen consumption and lower sensitivity to Seahorse compounds. In addition, control samples kept, which are

not kept on ice or are kept out of storage conditions for too long (4+ h), may exhibit a decay in their oxygen consumption activities.

## Planning and plate layouts

Large sample sets with hundreds of samples may require upward of 50 plates to assess ETC Complex I, Complex II, and Complex IV. With this in mind, high-throughput respirometry applications require a more nuanced and organized plate layout strategy. As discussed in the "Controls" section above, an ideal plate layout requires multiple control samples spread around the plate alongside experimental samples. In addition, an ideal plate layout

**Sample Plate**

| | 1 | 2 | 3 | 4 | 5 | 6 | 7 | 8 | 9 | 10 | 11 | 12 |
|---|---|---|---|---|---|---|---|---|---|---|---|---|
| A | 1 | 9 | 17 | 25 | 33 | 41 | 49 | 57 | 65 | 73 | 81 | 89 |
| B | 2 | 10 | 18 | 26 | 34 | 42 | 50 | 58 | 66 | 74 | 82 | 90 |
| C | 3 | 11 | 19 | 27 | 35 | 43 | 51 | 59 | 67 | 75 | 83 | 91 |
| D | 4 | 12 | 20 | 28 | 36 | 44 | 52 | 60 | 68 | 76 | 84 | 92 |
| E | 5 | 13 | 21 | 29 | 37 | 45 | 53 | 61 | 69 | 77 | 85 | 93 |
| F | 6 | 14 | 22 | 30 | 38 | 46 | 54 | 62 | 70 | 78 | 86 | 94 |
| G | 7 | 15 | 23 | 31 | 39 | 47 | 55 | 63 | 71 | 79 | 87 | 95 |
| H | 8 | 16 | 24 | 32 | 40 | 48 | 56 | 64 | 72 | 80 | 88 | 96 |

**Color Legend**

| |
|---|
| High-Activity Sample Controls |
| Medium-Activity Sample Controls |
| Low-Activity Sample Controls |
| Mouse Liver Mitochondria Seahorse Controls |
| Background Wells |

**NADH — Seahorse Plate 1**

| | 1 | 2 | 3 | 4 | 5 | 6 | 7 | 8 | 9 | 10 | 11 | 12 |
|---|---|---|---|---|---|---|---|---|---|---|---|---|
| A | | 9 | 17 | 25 | 33 | 41 | 9 | 17 | 25 | 33 | 41 | |
| B | 1 | 10 | 18 | 26 | 34 | 42 | 10 | 18 | 26 | 34 | 42 | 89 |
| C | LM | 11 | 19 | 27 | 35 | 43 | 11 | 19 | 27 | 35 | 43 | |
| D | LM | 12 | 20 | 28 | 36 | 44 | 12 | 20 | 28 | 36 | 44 | |
| E | | 13 | 21 | 29 | 37 | 45 | 13 | 21 | 29 | 37 | 45 | LM |
| F | 8 | 14 | 22 | 30 | 38 | 46 | 14 | 22 | 30 | 38 | 46 | LM |
| G | | 15 | 23 | 31 | 39 | 47 | 15 | 23 | 31 | 39 | 47 | 96 |
| H | | 16 | 24 | 32 | 40 | 48 | 16 | 24 | 32 | 40 | 48 | |

**NADH — Seahorse Plate 2**

| | 1 | 2 | 3 | 4 | 5 | 6 | 7 | 8 | 9 | 10 | 11 | 12 |
|---|---|---|---|---|---|---|---|---|---|---|---|---|
| A | | 49 | 57 | 65 | 73 | 81 | 49 | 57 | 65 | 73 | 81 | |
| B | 1 | 50 | 58 | 66 | 74 | 82 | 50 | 58 | 66 | 74 | 82 | 89 |
| C | LM | 51 | 59 | 67 | 75 | 83 | 51 | 59 | 67 | 75 | 83 | 44 |
| D | LM | 52 | 60 | 68 | 76 | 84 | 52 | 60 | 68 | 76 | 84 | |
| E | | 53 | 61 | 69 | 77 | 85 | 53 | 61 | 69 | 77 | 85 | LM |
| F | 8 | 54 | 62 | 70 | 78 | 86 | 54 | 62 | 70 | 78 | 86 | LM |
| G | 45 | 55 | 63 | 71 | 79 | 87 | 55 | 63 | 71 | 79 | 87 | 96 |
| H | | 56 | 64 | 72 | 80 | 88 | 56 | 64 | 72 | 80 | 88 | |

**SR — Seahorse Plate 3**

| | 1 | 2 | 3 | 4 | 5 | 6 | 7 | 8 | 9 | 10 | 11 | 12 |
|---|---|---|---|---|---|---|---|---|---|---|---|---|
| A | | 9 | 17 | 25 | 33 | 41 | 9 | 17 | 25 | 33 | 41 | |
| B | 1 | 10 | 18 | 26 | 34 | 42 | 10 | 18 | 26 | 34 | 42 | 89 |
| C | LM | 11 | 19 | 27 | 35 | 43 | 11 | 19 | 27 | 35 | 43 | |
| D | LM | 12 | 20 | 28 | 36 | 44 | 12 | 20 | 28 | 36 | 44 | |
| E | | 13 | 21 | 29 | 37 | 45 | 13 | 21 | 29 | 37 | 45 | LM |
| F | 8 | 14 | 22 | 30 | 38 | 46 | 14 | 22 | 30 | 38 | 46 | LM |
| G | | 15 | 23 | 31 | 39 | 47 | 15 | 23 | 31 | 39 | 47 | 96 |
| H | | 16 | 24 | 32 | 40 | 48 | 16 | 24 | 32 | 40 | 48 | |

**SR — Seahorse Plate 4**

| | 1 | 2 | 3 | 4 | 5 | 6 | 7 | 8 | 9 | 10 | 11 | 12 |
|---|---|---|---|---|---|---|---|---|---|---|---|---|
| A | | 49 | 57 | 65 | 73 | 81 | 49 | 57 | 65 | 73 | 81 | |
| B | 1 | 50 | 58 | 66 | 74 | 82 | 50 | 58 | 66 | 74 | 82 | 89 |
| C | LM | 51 | 59 | 67 | 75 | 83 | 51 | 59 | 67 | 75 | 83 | 44 |
| D | LM | 52 | 60 | 68 | 76 | 84 | 52 | 60 | 68 | 76 | 84 | |
| E | | 53 | 61 | 69 | 77 | 85 | 53 | 61 | 69 | 77 | 85 | LM |
| F | 8 | 54 | 62 | 70 | 78 | 86 | 54 | 62 | 70 | 78 | 86 | LM |
| G | 45 | 55 | 63 | 71 | 79 | 87 | 55 | 63 | 71 | 79 | 87 | 96 |
| H | | 56 | 64 | 72 | 80 | 88 | 56 | 64 | 72 | 80 | 88 | |

**NADH — Seahorse Plate 5** (columns 1–6); **SR** (columns 7–12)

| | 1 | 2 | 3 | 4 | 5 | 6 | 7 | 8 | 9 | 10 | 11 | 12 |
|---|---|---|---|---|---|---|---|---|---|---|---|---|
| A | | 1 | 89 | 1 | 89 | | | 1 | 89 | 1 | 89 | |
| B | 1 | 2 | 90 | 2 | 90 | 89 | 1 | 2 | 90 | 2 | 90 | 89 |
| C | LM | 3 | 91 | 3 | 91 | 44 | LM | 3 | 91 | 3 | 91 | 44 |
| D | LM | 4 | 92 | 4 | 92 | | LM | 4 | 92 | 4 | 92 | |
| E | | 5 | 93 | 5 | 93 | LM | | 5 | 93 | 5 | 93 | LM |
| F | 8 | 6 | 94 | 6 | 94 | LM | 8 | 6 | 94 | 6 | 94 | LM |
| G | 45 | 7 | 95 | 7 | 95 | 96 | 45 | 7 | 95 | 7 | 95 | 96 |
| H | | 8 | 96 | 8 | 96 | | | 8 | 96 | 8 | 96 | |

**Figure 2. Effective plate layout for high-throughput respirometry—transitioning from a single sample plate to five seahorse plates.**
To efficiently load study samples from their 96-well Sample Plate into Seahorse Plates, we have developed a setup and plate layouts, which satisfy the criteria required for large-scale studies and, importantly, will allow for the multichannel pipetting of your samples from the Sample Plate into the Seahorse Plate layouts. First, ensure that each Sample Plate, the 96-well plate containing all of your samples (top row), contains high-activity (green), medium-activity (purple), and low-activity (yellow) controls. Here, we have color-coordinated them and provided each sample of the Sample Plate with a number to map them as they are transferred to the Seahorse Plates. We propose splitting this one Sample Plate into five "Seahorse Plates" for respirometry. Seahorse Plates 1 and 2 are used for Complex I/Complex IV RIFS protocol, Plates 3 and 4 are used for Complex II/Complex IV RIFS protocol, and Plate 5 is half Complex I/Complex IV and half Complex II/Complex IV. Samples 9–48 from the Sample plate are run on Seahorse Plates 1 and 3 in duplicate, samples 49–88 are run on Seahorse Plates 2 and 4 in duplicate, and samples 1–8 and 89–96 are run on Seahorse Plate 5 in duplicate. Although Plates 1–4 are single-fuel plates, to save space Plate 5 is a two-fuel plate with half of the plate using NADH for Complex I and half of the plate using succinate/rotenone for Complex II. The control samples are manually added into columns 1 and 12 of each Seahorse Plate; if a control sample was included in the sample set of that Seahorse Plate (e.g., Sample 1, 8, 44, 45, 89, or 96 is a part of the sample set being run on that plate), then it is excluded from columns 1 or 12 because it is being run elsewhere on the plate. Using this plating format, all samples for 1 Sample Plate can be split into five Seahorse Plates such that they are run in duplicate and include all requisite study controls. (NADH = the post-NADH injection oxygen consumption rate is representative of a Complex I activity reading). (SR, succinate/rotenone; the post-succinate/rotenone injection oxygen consumption rate is representative of a Complex II activity reading).

should also allow for multichannel pipette loading of samples to make sample loading as streamlined as possible. Fig 2 depicts an example plate layout setup, which meets all of the criteria required for high-throughput respirometry processing.

For clarity in the terminology used in this study, plates containing study samples will be labeled as a "Sample Plate." Plates created for the purpose of loading into the Seahorse for a respirometry experiment will be referred to as a "Seahorse Plate."

For each 96-well sample plate, we add two low-activity, two medium-activity, and two high-activity samples in locations 1, 8, 44, 45, 89, and 96 (these locations represent the four corners and the center of each sample plate). From this 96-well sample plate, we split the samples into five Seahorse Plates. Seahorse Plates 1 and 2 satisfy the Complex I and Complex IV assessments of samples 9–88 in duplicate; Seahorse Plates 3 and 4 satisfy the Complex II

and Complex IV assessments of samples 9–88 in duplicate; and Seahorse Plate 5 satisfies Complex I, II, and IV assessments of samples 1–8 and 89–96 in duplicate. Note that columns 1 and 12 of each Seahorse Plate are reserved for control samples; low-, medium-, or high-activity controls are added to these columns if they are not within the sample set being assessed (e.g., Seahorse Plate 1 has Controls 44 and 45 within the sample set, so only Controls 1, 8, 89, and 96 are added to Columns 1 and 12), and positive mitochondrial controls are added to each plate, with the remaining wells acting as background controls for the Seahorse platform. Although this is not the only style of plate layouts sufficient for high-throughput respirometry, the provided example meets the main criteria for optimal study efficiency: all controls are included, the samples are run in duplicate, the samples can be loaded using a multichannel pipette, and the plate layouts are consistent for easy data handling at the end of the study.

# Analytical Considerations for High-throughput Respirometry

**Analytical considerations to consider for high-thoughput respirometry and their solutions.**

| Challenges to consider | Tools and parameters to consider using for this challenge | Solutions to consider for your study |
|---|---|---|
| How will you get your data from the instrument in an organized and consistent way? | Consider an automated spreadsheet. Seahorse data should be organized identically for all runs to allow for standardized data handling and analysis. | An automated spreadsheet (see Supplemental Data 1) can accomplish two important steps: (1) efficient transfer of data into an organized format, and (2) provide an organized template to apply quality control thresholds in an unbiased method. Automating a spreadsheet reduces the probability for errors and can streamline the data analysis process. |
| | Consider automated spreadsheets to consolidate multiple Seahorse runs into a consistent format. | An automated spreadsheet can collate multiple Seahorse into a single organized block of data for later quality control analysis. |
| | Consider providing each sample with a unique identifier to track its progress through the workflow and analysis. | Providing samples with an identifier, even as simple as "Plate Number"–"Plate coordinate" will allow for better sample tracking and will allow samples to be found easier for rerun selection and/or looking at individual sample performance in the study. |
| How can you automate quality control for large sample sets? What trends in the data can inform this quality control? | Look at random assortments of samples and observe any trends in your data. | Although data analysis should eventually be fully automated for large datasets, manually assess the quality of 10-20 samples. Determine whether there are any trends (i.e., OCR too high or low), which will need to be considered when creating data acceptance criteria or quality control metrics. |
| | Are your samples near the Seahorse lower limit of detection (insensitivity to compounds)? | Use a minimum OCR threshold to exclude samples near the lower limit of detection. Set the OCR threshold based on the performance of your low-activity control sample. Verify using smaller subsets of low-activity samples in your dataset, which could be manually assessed for accuracy. |
| | Are your samples near the Seahorse upper limit of detection (exhibiting J-curving)? | Samples near the upper limit of detection may exhibit depletion of oxygen from the well underestimating OCR. Although our automated spreadsheet does not have an accommodation for high-activity samples, rates over a certain threshold could be flagged for review. In addition, oxygen levels could be reviewed on the instrument or by using the differences between replicate measurements after substrate injections. |
| What acceptance criteria should I apply to the data? | How well did the control samples perform? Is your quality control procedure accurately eliminating poorly performing samples? | This step requires some manual assessment of smaller portions of your dataset. Common problems include samples having lower Complex IV capacity than Complex I/II capacity, samples running near the upper or lower limit of detection, poor sample reproducibility, and/or issues across an entire plate. Using subsections of the data to validate your quality metrics can be a way to confirm these samples should be excluded based on quality control criteria. In this way, you can fine-tune your analysis to match what your manual assessments. |
| | How many of your technical replicates per sample are performing properly? | We would recommend at least two acceptable traces for each ETC complex being measured, but in cases where samples spatially overlap or reruns may not be feasible, one trace may be acceptable for study results. |
| What samples should I rerun? | Which samples were near the limit of detection for the Seahorse? | Samples that do not pass quality control metrics can be rerun at higher sample volumes to bring them above the limit of detection of the respirometry platform being used. |
| | Which samples were inconsistent across technical replicates? | Certain subsets of data may be more inconsistent than others (often near the upper or lower limit of detection). Identifying and rerunning these subsets using the quality control techniques outlined will help determine which samples are performing poorly and how to best account for them in your study. |
| | How are you compiling and running these samples? | After compiling samples, you can generate new plate layouts to rerun these samples. Maintaining each sample's unique study identifier can ensure that the rerun data can be easily compiled with the original run data. |

In most Seahorse assays, analysis and quality control practices are typically done on a sample-by-sample basis. With these lower throughput experiments, an operator can look at each Seahorse trace individually to assess wells, which may not have injected properly, or technical replicates, which did not perform similarly and exclude them from the final analysis on this basis. With the hundreds of samples, looking at every sample and control is not feasible. Therefore, large sample sets demand a novel automated method for analysis and quality control, which can deal with the immense amount of data generated in high-throughput applications.

### Analysis of RIFS Seahorse data

The details for the analysis of Seahorse respirometry and RIFS can be found in Osto et al (2020); here, we cover the optimizations made to analyze large Seahorse datasets in a high-throughput manner. Three parameters must be measured for each sample: Complex I–driven respiratory capacity (NADH-stimulated OCR-antimycin A–stimulated OCR), Complex II–driven respiratory capacity (succinate/rotenone-stimulated OCR-antimycin A–stimulated OCR), and Complex IV–driven respiratory capacity (TMPD/ascorbate-stimulated OCR-azide-stimulated OCR). These values represent the ETC complex maximal respiratory activities, which can then be normalized to protein content to achieve a normalized maximal respiratory capacity with the units (pmol $O_2$/min)/μg protein. Fig 3A shows a representative Seahorse trace of Complex I and Complex IV activity.

### Automated data handling, calculations, and organization

To prevent possible user errors when handling and analyzing the Seahorse data and to keep the data in an organized format, an automated Excel worksheet was created to transform the normalized rate data coming from the Seahorse instrument into a universal format, which would be applied to all runs. This Excel worksheet has been included as a supplement to this study. In short, the bicinchoninic acid assay (BCA, total protein) normalization data for each sample were added to the Seahorse data using the normalization tools within Agilent Seahorse Wave Analysis software. From Wave software, each run was exported as an Excel document, which provides the run data in multiple sheets, one of which labeled "Normalized Rate (Plates)," and provides the normalized OCR of each well for every measurement cycle of the Seahorse. Measurement data could then be copied and pasted into the "Automated SH Plate Analysis" Excel worksheet where the data automatically fill into the accompanying tables and the following parameters are calculated: average baseline OCR, average first injection OCR, average antimycin A OCR, average TMPD OCR, average azide OCR, (average first injection OCR)-(average AA OCR), (average TMPD OCR)-(average azide OCR), and the difference between the two technical replicates for the injection OCR-AA OCR and TMPD OCR-azide OCR values. The average normalized OCR values consolidate the multiple measurements made after each injection. The injection OCR-AA OCR and TMPD OCR-azide OCR calculations

automatically determine the Complex I/II–driven respiratory capacity (depending on the injection used for that plate) and the Complex IV–driven respiratory capacity, respectively. In addition, these metrics are collated into two tables, which provide quick information regarding the performance of individual assay plates. These two tables can also be copied and pasted into the "Five-Plate to Column Analysis" worksheet, which consolidates the data from the five Seahorse runs of a single Sample Plate into eight columns that detail every calculated metric for each Seahorse trace. These tools allow each Seahorse run to be analyzed and organized in a uniform manner, simplifying data interpretation, troubleshooting, and analysis.

# Establishing Quality Control Parameters

High-throughput respirometry studies cannot afford to manually examine each individual data point to assess whether a sample should or should not be accepted for the study because of technical issues. We therefore needed to create a set of parameters to detect potential outliers or technical failures in raw Seahorse data. This section will detail and justify parameters we developed for unbiased, high-throughput respirometry data analysis. In this section, we validate our novel quality control methods using data generated during a high-throughput RIFS respirometry study of the human brain consisting of 703 blinded samples of either gray, white, or mixed brain matter (Mosharov et al, 2025). This represents a respirometry dataset of nearly 14,000 data points where we could test and validate different methods of quality control. The following section details and justifies these parameters and methods for unbiased, high-throughput respirometry data analysis as determined before, during, and after validation using this dataset.

### Oxygen consumption rate thresholding—low OCR threshold

As described in "Normalization to Tissue Content" section above, although loading all samples at equivalent volume for the Seahorse runs is a feasible method to achieve higher throughput, it does mean that lower OCR samples would be near the limit of detection of the Seahorse platform. With this challenge in mind, we decided on inclusion criteria requiring a minimum OCR threshold per μg of protein, which samples would have to meet. Samples that do not meet the threshold criteria may be set aside to be rerun at a larger volume (and therefore higher protein concentration).

Two orthogonal methods were used to determine a minimum OCR threshold for the dataset; a flowchart of the decisions made in establishing these OCR threshold values is outlined in Fig 4A. We began by manually examining every control sample trace to subjectively determine whether the control sample performed reliably on the Seahorse. In addition, any negative OCR values were excluded as negative values should not be possible in our setup and are a result of being within the noise of the instrument's measurements. Using Complex I controls as an example, the average of all control samples resulted in an average OCR of 0.63 with

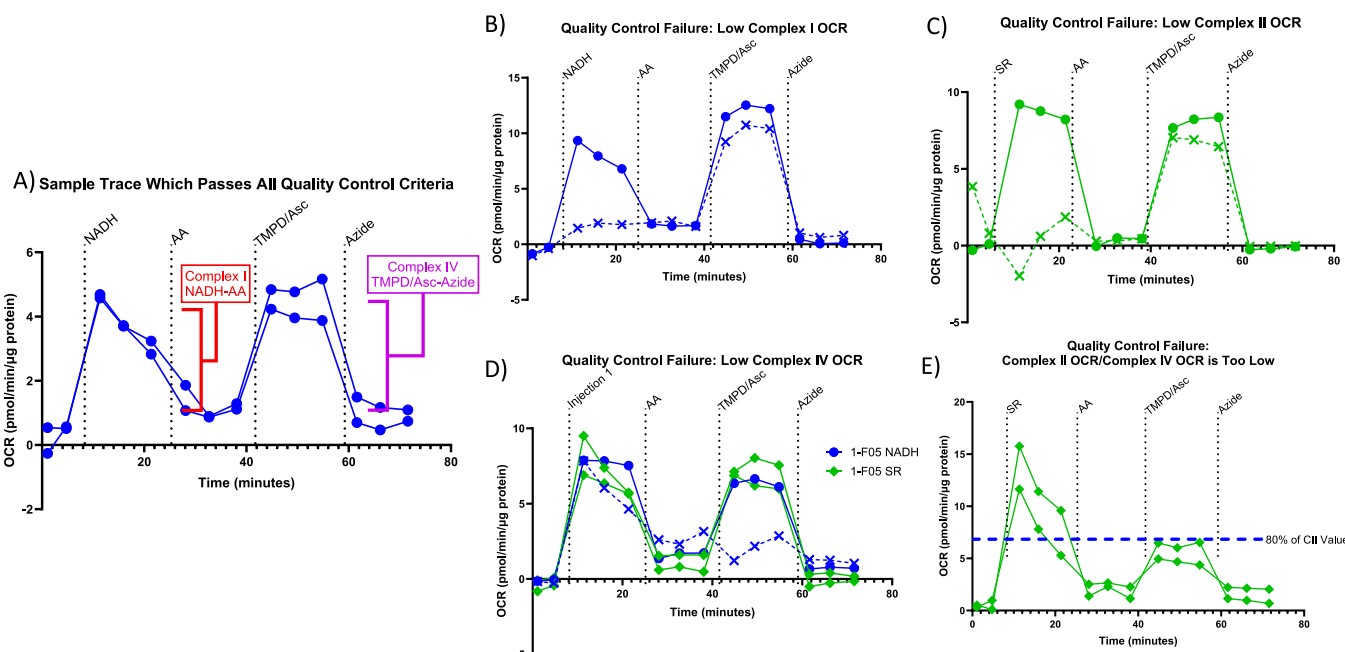

**Figure 3. Example of analytical quality control thresholds.**
This figure displays representative Seahorse traces from our validation study, which illustrate the types of samples that were excluded because they failed to pass our study's analytical quality control thresholds. In panels (B) through (D), the excluded trace is that with the dotted line and "X" markers. In E, both traces were excluded. **(A)** Seahorse traces from a brain sample that passed all quality control criteria. **(B)** Seahorse traces from a brain sample that failed Complex I oxygen consumption rate (OCR) threshold, with one trace exhibiting a Complex I value less than 0.5 pmol/min/μg. **(C)** Seahorse traces of a sample that failed Complex II OCR threshold, with one trace exhibiting Complex II value less than 0.5 pmol/min/μg. **(D)** Seahorse traces of a sample that failed Complex IV OCR threshold, with one trace exhibiting a Complex IV value less than 1.5 pmol/min/μg. **(E)** Seahorse traces from a brain sample that failed TMPD accommodation threshold with both traces exhibiting Complex IV values less than 80% of the value of their respective Complex II values. (NADH = the post-NADH injection OCR is representative of a Complex I activity reading). (SR, succinate/rotenone; the post-succinate/rotenone injection OCR is representative of a Complex II activity reading. AA, antimycin A, inhibitor of Complex III. TMPD/Asc, TMPD + ascorbate; the TMPD/Asc injection OCR is representative of a Complex IV activity reading).

a SD of 0.56 before manual trace exclusion. After manual trace exclusion, we calculated the average OCR of 0.88 with a SD of 0.26 (Fig 4C).

Statistically, limit of detection of a sample is determined via one of the two methods: 3.3 SDs above a blank sample (at a 95% confidence interval) or 1.645 times the SD of a low concentration sample above blank measurements (The International Union of Pure and Applied Chemistry IUPAC, 1997; Armbruster & Terry, 2008). Because blank samples within the Seahorse are treated as background (and therefore an activity of 0 OCR/μg), we initially set our limit of detection to (0 + 1.645*[0.26]) or 0.43, with 0 representing the OCR of a blank well, 0.26 representing the SD of a low-activity sample, and 1.645 as a defined value to determine the limit of detection of an assay. Comparing this 0.43 threshold with the samples we had determined that did not perform reliably on the Seahorse, we found that this value was not selective enough, leaving some manually determined poor-performing samples in the dataset. Alongside our previous method, we decided to iteratively increase the quality control threshold until two criteria were met for all white matter control traces: our population of batch correction controls fell within the 95% confidence interval of expected SD as calculated by linear regression of the other brain matter controls used in the study, and our manually excluded samples were all excluded (Fig 4B).

The final empirically determined OCR thresholds for this dataset were as follows: Complex I and Complex II samples must be greater than 0.5 OCR/μg protein, and Complex IV samples must be greater than 1.5 OCR/μg protein. Fig 3B–D shows representative Seahorse traces, which did not pass the Complex I, Complex II, or Complex IV OCR threshold values and were therefore excluded or rerun based on the inclusion criteria described below.

Knowing that our volumetric sample loading protocol would result in low-activity samples running near the limit of detection of the assay and knowing that the samples in our study were not precious and could be rerun multiple times if necessary, we settled on a threshold which would select for more sample reruns than would be required by the traditional limit-of-detection definitions. Although this method to determine a low OCR threshold based on the limit of detection of the assay is adaptable to multiple types of studies, it must be mentioned that OCR normalized to ug protein will differ based on the sample type, sample source, and experimental conditions, and therefore, our specific thresholds will not be universally applicable to all samples. Pretesting experiments using a large batch of background wells and low-activity controls could better determine a limit of detection before beginning the study and could guide the methodologies and quality control metrics used throughout your assessment.

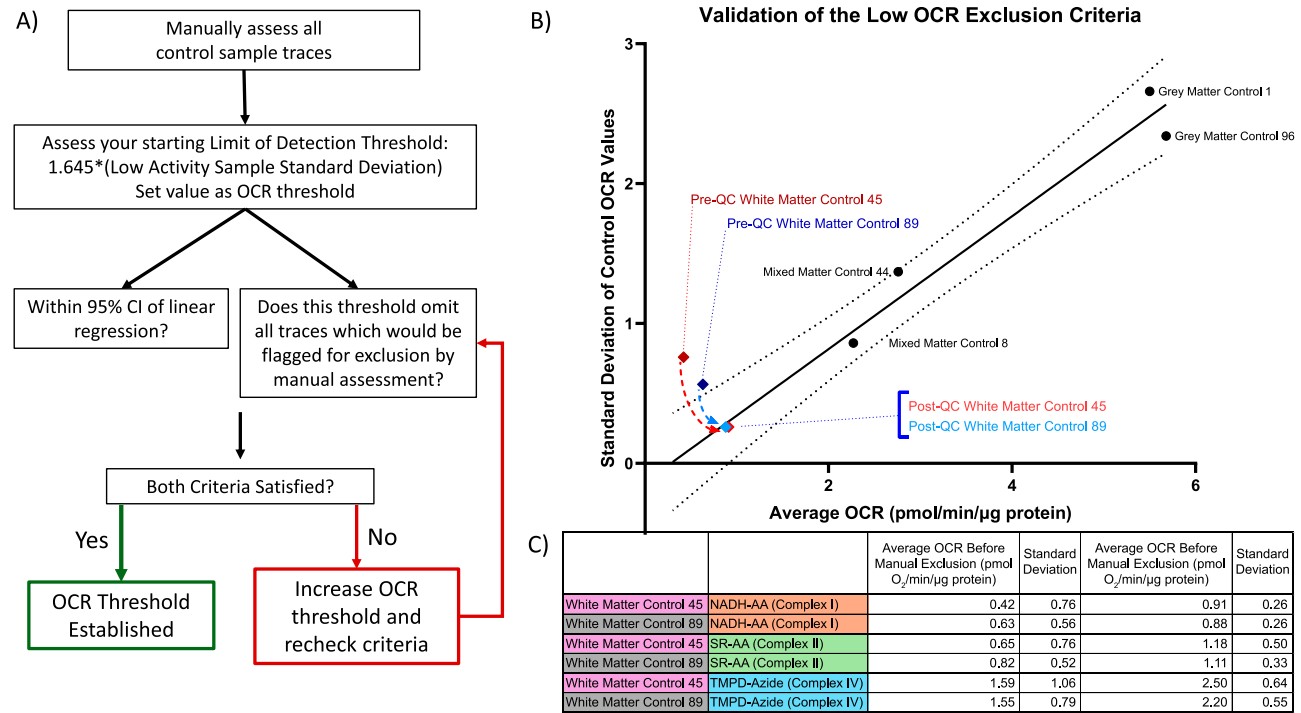

**Figure 4. Decision tree for establishing oxygen consumption quality control thresholds.**
**(A)** A flowchart of decision-making process when selecting an appropriate oxygen consumption rate (OCR) threshold. First, brain matter control samples were manually inspected to determine the appropriate signal-to-noise range of OCR detection. Next, the initial threshold was set to 1.645 times the SD of our low-activity control sample (0.43 pmol/min/µg), but increased iteratively to meet our criteria that the controls must be within the 95% linear regression of variance and must exclude all traces that would have been excluded by manual selection. The final OCR thresholds were 0.5 pmol/min/µg for Complex I and II and 1.5 pmol/min/µg for Complex IV. **(B)** Validation of the low OCR quality control threshold value established in (A) using Complex I brain matter control sample data. Variance (in the form of SD) is assessed across the brain controls compared with their average OCR levels. The filled line indicates the linear regression of the control samples (after thresholding), whereas the dotted line is the 95% confidence interval. Before threshold application (dark red—sample 45; and dark blue—sample 89), two white matter controls were outside of the 95% confidence interval. Adjustment using our final threshold value results in a SD and average OCR of the remaining control values, which results in these values now within the 95% confidence interval (light red—sample 45; and light blue—sample 89). **(C)** This table showcases the changes in average OCR and SD of the white matter brain controls before and after quality control using our low OCR threshold. **(A, B)** Although the Complex I data are showcased in (B), we include in this table the numerical changes associated with thresholding the data observed in Complex II and Complex IV using an identical method to (A).

## Oxygen consumption rate thresholding—TMPD accommodation

After initial analysis of the data and manual observations of assorted sample traces, we observed that certain samples would have valid Complex I or Complex II values, but the subsequent Complex IV measurement would be lower by comparison. Given the sequential nature of the Seahorse assay and the fact that stimulation of Complex I or II is measuring OCR of Complex IV, it should not be possible for a sample to have a higher OCR for Complex I or II than for Complex IV. However, there are several technical reasons this might occur: the TMPD/ascorbate (Complex IV substrate) may not have injected properly for these samples; there may be some sample material lifting from the plate during the time course of the Seahorse experiment; or, given that multiple samples that exhibited this phenomenon had higher OCR values, this may be a result of oxygen depletion from the Seahorse well (if depletion happens in too many samples, they should be rerun at a lower sample volume or should be diluted to be within the signal-to-noise range of the instrument). With these technical considerations in mind, we implemented an automated method of identifying these types of samples followed by either rerunning or

excluding them from the study. After looking through a selection of samples and gray matter loading controls, which exhibited this phenomenon, we determined that the Complex IV OCR value must be at least 80% of its respective Complex I or Complex II OCR value. Fig 3E shows an example Seahorse trace, which did not meet the TMPD accommodation criteria.

## Study inclusion criteria and management of sample reruns

With the two quality control measures in place (low OCR threshold and TMPD accommodation), each trace underwent automated analysis individually from its technical duplicate. Each trace was first analyzed via the low OCR threshold and provided with a "Pass" or "Fail" for Complexes I, II, and IV. At this point, the passes and fails were tallied; if a sample had at least one passing Complex I measurement, one passing Complex II measurement, and two passing Complex IV measurements, then the sample was accepted to the study and did not require a rerun. Any sample that did not meet these criteria was placed into one of four categories depending on what quality control criteria did not pass: only Complex I failure, only Complex II failure, Complex I and Complex II

Failure, or only Complex IV failure. These samples were then accordingly binned for future reruns where only Complex I and only Complex II failures would only need to be rerun for their respective failing complex, only Complex IV failures would be run for Complex I/IV because Complex I values are typically more prone to failure and would benefit more for additional replicates, and Complex I and Complex II failures would have to be rerun for both complexes.

Thresholded samples were manually assigned to a Rerun Sample Plate Layout and run (as discussed previously) at 25 µl rather than 15 µl of sample volume. These samples were analyzed using a new low OCR threshold of 0.3 pmol $O_2$/µg protein determined identically to the threshold discussed in the "Oxygen consumption rate thresholding—low OCR threshold" section. If the rerun results increased the number of successful measurements such that a sample now meets the acceptance criteria for the Complex I, II, and/or IV measurement, this sample was included in our study.

An example of the quality control routine is shown in the Excel worksheet entitled "Plate 4 Data QC Filtering." In this worksheet, all samples from a single Sample Plate are analyzed as described above. First, Seahorse readouts are marked for whether they pass the OCR threshold criteria. If they pass this criterion, they are then assessed to determine whether they pass the TMPD accommodation threshold. If both of these thresholds are met, the sample is not rerun and is accepted for the study. However, if a sample does not pass our study inclusion criteria for either Complex I, Complex II, or Complex IV, then they are rerun using the appropriate assay and the rerun data are added into the study set using each sample's unique study identifier. Samples that did not pass the acceptance criteria after a rerun using a higher volume of homogenate are excluded from the final dataset. Of the 703 original brain samples, 70/35/19 did not have acceptable readouts for Complex I, II, and IV, respectively.

## Discussion

In this study, we describe the challenges associated with performing respirometry in high-throughput and large dataset applications and the approaches that can be used to streamline these types of experiments. Although we have attempted to make the approaches presented universally applicable, some of the choices made will need to be tailored for your specific datasets and/or applications. For example, our automated analysis sheets are designed to work with the mix/measure protocol used specifically in this study. Although our spreadsheets could be easily adapted to other experimental settings, these adjustments would still have to be made to be adapted to your study. This becomes especially important with regard to the acceptance criteria, which were largely made based on the trends that we observed after running and analyzing our control samples. Although we have outlined the trends in our dataset and the analysis choices we made in response, we recommend that researchers carefully review data, particularly initial testing subsets, for the potential signs, which are indicative of suboptimal respirometry data (e.g., the troubleshooting sections of Osto et al [2020] and Divakaruni

and Martin [2022]) that can be addressed early. We recommend observing a random assortment of data points to pinpoint if any issues exist and if so, how to handle these events in an unbiased way. In addition, others have proposed their own protocols for medium-throughput respirometry protocols (Underwood et al, 2020) and generalized respirometry statistical analysis protocols (Yépez et al, 2018), which are valuable resources to pair alongside our approaches to make your study optimized and bespoke to your needs. Furthermore, although we have focused on respirometry in previously frozen specimen in this study for its ability to collect, store, and process samples in large quantities, the principles outlined in this study can be adapted to fresh specimen where oligomycin-insensitive respiratory measurements and coupled versus uncoupled respiration measurements could be optimized and collected in the same methods as described here.

To the discussion on observing potential signs of suboptimal respirometry data, in our validation study we chose not to include analysis and/or quality control criteria to eliminate depletion of oxygen in the well, a Seahorse respirometry artifact colloquially known as "J-curving." Discussed in greater detail in Osto et al (2020), oxygen depletion (J-curving) occurs when excessive oxygen consumption results in anoxic oxygen levels within a well. When this happens, OCR measurements become no longer linear, preventing the Seahorse instrument from being able to accurately determine OCR through the algorithm calculation of the slope of oxygen partial pressure over time (Fig S1A). Although, in our assessments, we observed very few samples that exhibited this phenomenon (our samples trended toward having lower OCR levels rather than higher OCR levels based on our pretesting experiments), we acknowledge that this may not be the case for all studies and/or sample types. Multiple accommodations can be made to account for samples that exhibit this artifact in a high-throughput method, both accounting for a change in linearity during the measurement of oxygen consumption over time: (1) rerun or exclude any sample with a rate over 500 pmol/min before normalization as per Divakaruni and Martin (2022); (2) an assessment of the $R^2$ (square of the linear correlation coefficient) for each measurement cycle would allow for outlier detection of abnormally low $R^2$ values indicative of a deviation from linearity caused by J-curving (Fig S1B); and (3) alternatively, one may calculate the slope of oxygen pressure over time for the first 3–4 points of each measurement versus the slope of the last 3–4 points of the measurement. A deviation in slope determined via this method could be used as a quality control marker to identify J-curving measurements/samples (Fig S1C). Analyzing your Seahorse traces using the "point-to-point" analysis mode may also make any signs of oxygen depletion (J-curving) more clear, in cases where visual analysis may be difficult. Protocol adaptations like shortening measurement times can reduce oxygen depletion symptoms, but could in turn alter the sensitivity of the instrument in low-activity samples. In a scenario where your samples are exhibiting this oxygen depletion (J-curve) phenomenon, there are multiple methods that could easily be incorporated into your high-throughput respirometry workflow using the data which are already provided by the Seahorse instrument.

Our laboratory has the benefit of having multiple Seahorse XF96 instruments, which can be used simultaneously to reduce the

number of freeze–thaw cycles on the samples in high-throughput applications. For most labs equipped with a single instrument, extra considerations may need to be made to the number of samples that can be run. In addition, one should take into consideration sample integrity throughout the study and approaches to minimize sample degradation of ETC complexes throughout the course of testing. We show throughout our study, though, that there was no significant difference between control samples run on any of the four Seahorse instruments used in our study (Fig S2).

In our validation study, we chose to run our samples in duplicate. Although most protocols would recommend a minimum of three technical replicates per sample and more would be optimal, the scope of your study must be considered. Adding a third replicate to a study, this size of our validation study would ultimately increase the workload by hundreds of additional wells. We made the decision to run in duplicate and develop the more inclusive sample rerun criteria to run samples a third or fourth time when needed to account for the lack of additional replication on first run of the samples. There could be studies where additional replicates would be the more rational decision, like samples that are very sensitive to freeze–thaw cycles and could not be rerun as regularly, or samples that during pretesting are not as homogeneous in activity and would need more replicates to gain assurance in your oxygen consumption data.

Although in our validation study we chose to include a limit-of-detection (low OCR) threshold, and a Complex IV threshold to quality-control our dataset, this is certainly not the only types of quality control that could be applied or may need to be applied to achieve consistent and reliable data. For studies that have larger replicate numbers, outlier analyses can help identify poorly performing samples using analyses like median absolute deviation (Yépez et al, 2018) or a set SD from the mean to automatically identify rerun samples. It is important to consider that different studies will require different quality control techniques, and depending on the trends observed in your data, there will be some combination of techniques that will ensure that your data are reliable and that any samples that are suboptimal are either removed or set aside to be run again. Although strictly mathematical assessments may work, we feel that it is of great importance to use a set of controls or a small sample set to iterate upon purely mathematical quality control methods to ensure that they are filtering samples properly and that your methods are suitable to the trends of suboptimal performance, which may exist in your dataset. It is important to note that although we have outlined multiple techniques to quality-control your dataset, our specific quality control metrics are bespoke to our study and do not represent a rule to be applied to all respirometry tissues and studies.

Although reproducibility of your dataset is always of great importance when performing and analyzing large datasets, it must be noted that sample variance in respirometry can differ greatly depending on the sample type and the experimental setup. For example, brain samples like that used in our validation study can exhibit larger variation in OCRs as a result of fattier white matter making precise pipetting more difficult, especially compared with more uniform and easy-to-homogenize samples like liver. With this in mind, using an aggregate marker for reproducibility like the average of multiple controls can help track run-to-run reproducibility (Fig S3). Furthermore, as showcased in Yépez et al's study,

as oxygen consumption increases, so too does the SD of your respirometry dataset (Yépez et al, 2018). So the assessment of percent coefficient of variation or other reproducibility metrics relying on SD may not be as reliable if your dataset exhibits a range of respirometry activity.

One of the most important notes in a study of this size is to prepare, test, and practice the protocols as much as possible before running the entire sample set. From our experience, we found that many of the considerations that became vital to the success of a study, like loading by volume rather than protein concentration and finding a plate map that allowed all samples to be loaded using a multichannel pipette, were first identified during the initial pretesting experiments. Assuming that the sample material is not limited or precious, we strongly advise performing some small-scale experiments and analyses before expanding to a sample set of hundreds of samples.

Ultimately, as respirometry becomes more accessible in the mitochondrial research field, we will soon need the resources and expertise to perform and analyze larger and larger datasets. Before high-throughput analytical instruments and analysis tools become available, we believe the approaches outlined will provide researchers with the experimental considerations, procedural knowledge, and analytical framework to execute larger scale respirometry studies with increased efficiency.

## Data Availability

In this study, references are made to Excel worksheets containing automated calculation sheets and example data from our validation study. These are included as supplemental material to this study in the form of an Excel notebook with appropriately labeled worksheets. All data for our validation study are available upon request.

## Supplementary Information

## Acknowledgements

We would like to acknowledge our funding source from the following NIH grants: R01DK144611, 5R01DK141923, R01DK078760, 1P01HL170952, R01CA208642, and 5R01DK107397. We would also like to acknowledge the extended members of the laboratory of O Shirihai and the laboratory of M Picard for their input, recommendations, and advice throughout the development of these methodologies. In addition, we acknowledge the contributions of the many institutions and teams responsible for their part in the initial MitoBrainMap study for their part in the study.

### Author Contributions

CA Osto: conceptualization, data curation, formal analysis, validation, investigation, visualization, methodology, project administration, and writing—original draft, review, and editing.

EV Mosharov: conceptualization, resources, data curation, formal analysis, supervision, funding acquisition, investigation, visualization, methodology, project administration, and writing—review and editing.

AM Rosenberg: conceptualization, resources, data curation, supervision, investigation, visualization, project administration, and writing—review and editing.

M Picard: conceptualization, resources, supervision, funding acquisition, investigation, visualization, project administration, and writing—review and editing.

L Stiles: conceptualization, resources, data curation, formal analysis, supervision, investigation, visualization, methodology, project administration, and writing—review and editing.

O Shirihai: conceptualization, resources, data curation, supervision, funding acquisition, investigation, visualization, project administration, and writing—review and editing.

## Conflict of Interest Statement

The authors declare that they have no conflict of interest.

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
