## [Reviewer comments · Life Science Alliance]

Scalable Workflows for High-Throughput Respirometry

Corey Osto, Eugene Mosharov, Ayelet Rosenberg, Martin Picard, Linsey Stiles, and Orian Shirihai
DOI: <https://doi.org/10.26508/lsa.202503446>

Corresponding author(s): Orian Shirihai, University of California, Los Angeles

Review Timeline:	Submission Date:	2025-07-08
	Editorial Decision:	2025-07-21
	Revision Received:	2025-10-03
	Editorial Decision:	2025-11-03
	Revision Received:	2026-01-23
	Editorial Decision:	2026-02-19
	Revision Received:	2026-02-24
	Accepted:	2026-02-27

Scientific Editor: Tim Fessenden

Transaction Report:

July 21, 2025

Re: Life Science Alliance manuscript #LSA-2025-03446

Dr. Corey A Osto
University of California, Los Angeles
Molecular and Medical Pharmacology
10833 Le Conte Avenue, Los Angeles, CA 90095
CHS 23-100
Los Angeles, CA 90095

Dear Dr. Osto,

Thank you for submitting your manuscript entitled "Optimization of Respirometry Assays for Large-scale Studies" to Life Science Alliance. We have now assessed your paper and discussed it within our editorial team.

We appreciate these results describing novel approaches and technical considerations to enable high-throughput respirometry measurements. However the methodology described herein relates primarily to a prior publication in Nature and provides readers with expanded details on the methods used in that particular research article. This limits the broad appeal and utility of the method for LSA readers. Owing to this central limitation, the editors felt that the study would be a poor fit for LSA. We have thus decided not to subject your manuscript to a lengthy external review process.

This work is well-suited for publication in Bio-Protocol, which explicitly publishes methodologies linked to previously published research papers. See <https://bio-protocol.org/en/about?type=aims>. We strongly recommend considering this outlet for your work, if you have not already.

Because of our interest in this general topic and the potential utility of high-throughput metabolic studies using respirometry, we would be open to considering a significantly revised methods paper. A suitably revised manuscript must contain primary data as a proof-of-principle and offer readers a broader, general approach to this method that extends beyond the related paper already published.

I am sorry that our answer on this occasion is not more positive, and I hope that this outcome will not dissuade you from submitting other manuscripts to us in the future.

Thank you for your interest in Life Science Alliance.

With kind regards,

September 25, 2025

Dear Tim Fessenden,

The authors of manuscript #LSA-2025-03446 have requested an appeal. Their comments are below.

Dear Life Science Alliance Editing Team,

Thank you for your feedback on our recent submission. You stated that Life Science Alliance would be interested in the general topic of high-throughput respirometry studies if we revised the previously submitted methods paper. We provide here a significantly revised manuscript which describes a novel approach we have developed to allow for high-throughput, high sample number assessments of mitochondrial function via respirometry. The manuscript details the considerations taken in the development of these approaches, placing an emphasis on how these methodologies can be adapted to samples sets for respirometry of any size or sample type. We include examples and primary data showcasing the challenges and pitfalls associated with high-throughput and large dataset respirometry. Additionally, we use a primary dataset to validate our automated data analysis and quality control methods to ensure that they will be applicable to future studies. We hope that you will take this revised manuscript into consideration for your journal.

Thank you,
Corey Osto, PhD
Linsey Stiles, PhD
Orian Shirihai, PhD

You can accept or decline this request from the manuscript using the following link:

<https://lsa.msubmit.net/cgi-bin/main.plex?el=A7Na7BeL5A5Clwv2F5A9fdLMvnXezkvQtSzxNQr5N1eQZ>

Sincerely,

Editorial Staff

October 3, 2025

MS: LSA-2025-03446

Dr. Corey A Osto
University of California, Los Angeles
Molecular and Medical Pharmacology
10833 Le Conte Avenue, Los Angeles, CA 90095
CHS 23-100
Los Angeles, CA 90095

Dear Dr. Osto,

Our prior decision on your manuscript entitled "Optimization of Respirometry Assays for Large-scale Studies" has now been reconsidered. We appreciate the expanded and revised methodological considerations for high throughput respirometry, supported by an analysis of primary data. I am pleased to let you know that we have decided to consider the revised manuscript at LSA and will send this for peer review.

Please use the following link to submit your revised manuscript. Only after receipt of the revised manuscript will we begin contacting reviewers.

<https://lsa.msubmit.net/cgi-bin/main.plex?el=A6Na7BeL1A7CuMt6I1B9ftdwo08ZMxLZJuxbW5z561QZ>

Yours sincerely,

November 3, 2025

Re: Life Science Alliance manuscript #LSA-2025-03446R-A

Dr. Corey A Osto
University of California, Los Angeles
Molecular and Medical Pharmacology
10833 Le Conte Avenue, Los Angeles, CA 90095
CHS 23-100
Los Angeles, CA 90095

Dear Dr. Osto,

Thank you for submitting your manuscript entitled "Novel Optimizations Allow for High-throughput Respirometry" to Life Science Alliance. The manuscript was assessed by expert reviewers, whose comments are appended to this letter.

As you will see, reviewers are in broad agreement that this method will be valuable for high-throughput metabolism/respirometry studies. Reviewers 1 and 2 each made important and partially overlapping suggestions to improve this work which should be addressed in a revised manuscript. Both reviewers felt this work must provide a quantitative approach to detecting and filtering curves that indicate out-of-range function. Reviewer 2 similarly requested a quantitative approach to detecting and reporting batch effects.

I would be happy to discuss the revision in more detail via email or phone/videoconferencing if helpful.

While you are revising your manuscript, please also attend to the below editorial points to help expedite the publication of your manuscript. Please direct any editorial questions to the journal office. When submitting the revision, please include a letter addressing the reviewers' comments point by point.

Thank you for this interesting contribution to Life Science Alliance. We hope that the comments below will prove constructive as your work progresses, and we are looking forward to receiving your revised manuscript.

Sincerely,

B. MANUSCRIPT ORGANIZATION AND FORMATTING:

Reviewer #1 (Comments to the Authors (Required)):

This manuscript describes a methodology to facilitate bioenergetics measurements and analysis in up to 10k+ samples (using the 96-well format Seahorse analyzer). This piece is well organized, clearly written, covers most key aspects, and is expected to be useful, even for experiments with far fewer samples.

I have only a few points that I think are important to add:

1. Major point concerning the "J curve" problem. I understand that manual checking raw traces for each sample is impractical when there are many samples. However, the J curve issue can result in false data, and, eventually, false conclusions, and therefore is a serious issue. In particular, I think the issue needs more attention than it has been given - in the text, and in the workflow.

First, I think a non-linear J curve O₂ trace needs to be better described in the text, so that people better understand what exactly this is, and that there can be major consequences. This can be done by simply indicating that the algorithm to compute OCR assumes linearity of the O₂ trace, departures from linearity can compromise the calculation and that, in practice, departures from linearity most often appear as a J (initial linearity followed by rapid tapering off of the slope sometimes all the way to a horizontal line). Then it should be stated that the OCR calculated from such a trace will underestimate the linear portion of the O₂ trace, sometimes greatly underestimating it. A problem is that the underestimation may be such that, what is actually a high rate, in fact shows up as a moderate rate; in this case, the sample might not be flagged as one to look into. Another point is that J curving is not necessarily associated with a signal that is too high (and, thus, decreasing sample size does not necessarily remediate).

Highlight - by circling - an example of a J curve in Figure 1C - the 2nd TMPD/ascorbate trace shows it to some extent, but it would be better if you could provide a more obvious example.

While manual inspection of the traces may not be practical, I think that an additional level of QC needs to be included to look out for the problem. A. Regardless of sample numbers, I think readers should be encouraged to at least be aware that they can view the raw O₂ traces and that those traces are in fact the first stop of the QC process. B. If sample numbers are not too high, then manual inspection of raw O₂ traces is feasible and should be done. C. If sample numbers make manual inspection impractical, then there can be a few approaches that can be imagined, requiring more or less effort. One is to manually spot-check representative wells from different types of samples.

Another idea is to generate a script to re-organize the raw O₂ data to be able to analyze it in a way that compares the initial slope (first 4-5 points) and the slope calculated from the last 4-5 points, then determining, likely with the help of roughly linear data, if the difference or ratio of the two slopes is outside of some range. What would be great is if Agilent would agree to reformatting the raw O₂ data such that the output file from each experiment provided the raw O₂ value time course per well. Maybe Agilent could be motivated to provide that if it's made clear to them that it would be useful for screening purposes? Perhaps a script that re-orders those data could be provided as a plug-in, and maybe they could charge for it, to induce them to do this.

2. Regarding whether samples are within signal, the oligomycin-insensitive and uncoupler-driven rates are not really discussed (I realize that these tools were not emphasized - but many people may want to use them). I think you need to add a statement or two along the lines that, if one or both of those conditions are important, then those conditions also need to be considered during the optimization phase. It might even be mentioned that, for samples with large O₂ consumption scope, running two different amounts of the sample might be needed of both oligo and uncoupler rates are of interest.

3. Concerning very low O₂ consumption rates, how this appears on the raw O₂ trace should be stated - namely, that it appears as a downward slope that is barely indistinguishable from noise. Also, would it be useful to mention that prolonging the read can increase the reliability of the measurement when the OCR is low.
4. Concerning the use of liver mitochondria as a tool for plate to plate comparison, this is a good idea, but it would be useful to remind readers that coupled respiration of liver mitos decays with time on ice; this can be an important consideration for those having access to 1 or 2 SH instruments.
5. That 2 technical replications can be used to almost implied by the description of the workflow. Sometimes more than 2 technical reps are needed. I think readers should be reminded of this towards to the end of the manuscript.
6. Top of p11: please provide units of OCR in the text.
7. Figure 2: the color coding is not clear to me. Please improve the explanation in both the figure and the legend.
8. Figure 4A: the statement "All traces assessed for manual exclusion would be excluded?" is not clear to me.

Reviewer #2 (Comments to the Authors (Required)):

The paper presents a workflow for large-scale Seahorse respirometry, including single-volume lysate loading, automated quality control, and optimized plate layouts. The authors provide practical guidance and tools to streamline the handling and analysis of hundreds of samples. The main advance is procedural, offering a framework to improve efficiency and consistency in high-throughput mitochondrial studies of heterogeneous tissue.

Limitations / concerns

- Normalization issue: Single-volume lysate loading does not account for differences in protein or mitochondrial content between tissue types.
- Reproducibility not shown: No inter-plate (repeatability) or intra-plate (CV) data to confirm method consistency. Would be interesting to see the assay performed on same samples in a different day. Considering the large number of samples available it would seem quite possible to plot inter-plate (different dates, same sample) Bland-Altman analysis seems possible. Similarly, it would be useful to see intra-plate coefficient of variation (%) from technical repeats of the same sample in the same plate.
- Manual trace exclusion unclear: Criteria for removing traces are subjective and not fully described. I would suggest a mathematical method. For identifying outlier wells in Seahorse assays, a robust approach is to use the Median Absolute Deviation (MAD) across technical replicates and participants. Wells with Mi values exceeding a defined threshold (e.g., $M_i > 5$, Hoaglin 2013; Yopez et al. 2018) can be excluded. This provides an objective, reproducible method for outlier removal rather than relying on subjective manual trace exclusion
- "High-throughput": While the authors report using multiple Seahorse XF96 instruments in parallel to reduce freeze-thaw cycles, it is unclear whether the same sample produces consistent results across different machines. Moreover, while this approach may be practical for their lab, most laboratories will only have access to a single Seahorse, limiting the general relevance of this 'high-throughput' claim. Including inter-instrument reproducibility data and clarifying the scalability limitations would strengthen the manuscript.
- No comparison to prior work: The method does not cite or acknowledge established approaches (e.g., Underwood et al., 2020).
- QC thresholds not validated: Acceptance criteria were empirically chosen from their dataset, without independent validation.
- Potential for bias: Fixed-volume loading could over- or under-represent OCR in different tissue types, affecting biological interpretation.

If the authors provide additional validation demonstrating intra-plate and inter-plate reproducibility, inter-instrument consistency, and the relationship between lysate volume, protein content, and OCR, along with an objective, standardized method for outlier removal (e.g., MAD-based M_i statistic), the robustness and generalizability of their workflow would be greatly strengthened.

'Referee Cross-Comments' : no comments

Reviewer #3 (Comments to the Authors (Required)):

This article discusses novel methodologies, approaches and considerations to scale mitochondrial respirometry studies to hundreds to thousands of samples. Furthermore, the authors validated their high-throughput applications with 703 human brain samples.

This is an important and timely study helpful to any researchers studying large numbers of samples in diverse diseases and pathologies.

01/23/2026

Response to Reviewer Comments: Life Science Alliance manuscript #LSA-2025-03446R-A

To the reviewers and Life Science Alliance Editing Team,

Thank you for your time and effort in reviewing our manuscript entitled "Novel Optimizations Allow for High-throughput Respirometry". We appreciate the encouraging comments regarding the valuable nature of our manuscript and we have taken the measures to address the comments of the reviewers to improve our manuscript.

Our responses to the reviewers' comments are as follows:

Reviewer 1:

- 1) Reviewer 1 states: "Major point concerning the "J curve" problem. I understand that manual checking raw traces for each sample is impractical when there are many samples. However, the J curve issue can be result in false data, and, eventually, false conclusions, and therefore is a serious issue. In particular, I think the issue needs more attention than it has been given - in the text, and in the workflow...."
- We agree with reviewer 1 that J-curving can be an issue within Seahorse data sets which could result in false data and/or false conclusions from the data. As suggested we addressed this comment in the revised manuscript with the following edits. We clarified the definition and provided a more detailed description of the J-curving phenomenon both within the text (The 'Sample Loading with Respect to Normalization of Tissue Content/Mitochondrial Activity' section of the manuscript), and we added a new paragraph in the Discussion section explicitly describing J-curving, providing a new supplemental figure with a better example of what J-curving looks like, and including within this Supplemental figure two potential methods to detect J-curving in a quality control workflow. This quality control assessment includes the reviewer's suggestions of (a) comparing the difference in slope between the first 4 and last 4 oxygen partial pressure measurements, and (b) comparing linearity via r^2 linear regression analysis to determine whether a sample is or is not exhibiting J-curving.
- 2) Reviewer 1 states: "Regarding whether samples are within signal, the oligomycin-insensitive and uncoupler-driven rates are not really discussed (I realize that these tools were not emphasized - but many people may want to use them). I think you need to add a statement or two along the lines that, if one or both of those conditions are important, then those conditions also need to be considered during the optimization phase. It might even be mentioned that, for samples with large O₂ consumption scope, running two different amounts of the sample might be needed of both oligo and uncoupler rates are of interest."
- Our protocol emphasizes utilizing previously-frozen samples for respirometry, which allows for samples to be collected and stored on a larger scale. However, frozen samples can only measure uncoupled respiration and not ATP synthesis capacity. We understand that this may not be the case for all users and we have included a sentence in the discussion in case the end user wants to perform analyses in coupled respiration in their samples.
- 3) Reviewer 1 states: "Concerning very low O₂ consumption rates, how this appears on the raw O₂ trace should be stated - namely, that it appears as a downward slope that is barely indistinguishable from noise. Also, would it be useful to mention that prolonging the read can increase the reliability of the measurement when the OCR is low."

- Per the reviewer's suggestion, we have more directly defined what low oxygen consumption (near the limit of detection) looks like in the Seahorse platform.
- 4) Reviewer 1 states: "Concerning the use of liver mitochondria as a tool for plate to plate comparison, this is a good idea, but it would be useful to remind readers that coupled respiration of liver mitochondria decays with time on ice; this can be an important consideration for those having access to 1 or 2 SH instruments."
- As described in our response to point 2 by Reviewer 1, we have focused this paper on previously-frozen samples, a method that does not allow for the analysis of coupled respiration. To address this comment, we have more directly stated that long periods of storage (even on ice) can result in a decay of oxygen consumption activity.
- 5) Reviewer 1 states: "That 2 technical replications can be used to almost implied by the description of the workflow. Sometimes more than 2 technical reps are needed. I think readers should be reminded of this towards to the end of the manuscript."
- As suggested, we have added a paragraph in the discussion to remind readers that different applications will potentially require more than two technical replicates. This paragraph details that there are multiple considerations and/or compromises to make when choosing how many technical replicates to include. This includes the size of the study, the ability to rerun samples, and the necessity for replication in the study.
- 6) Reviewer 1 states: "Top of p11: please provide units of OCR in the text."
- Units of OCR have been added throughout the text.
- 7) Reviewer 1 states: "Figure 2: the color coding is not clear to me. Please improve the explanation in both the figure and the legend."
- To address the reviewer comment we have clarified the coloration of Figure 2 both with altering the figure and altering the legend of Figure 2.
- 8) Reviewer 1 states: "Figure 4A: the statement "All traces assessed for manual exclusion would be excluded?" is not clear to me."
- The wording in Figure 4 has been altered to make it more clear.

Reviewer 2:

- 1) Reviewer 2 states: "Normalization issue: Single-volume lysate loading does not account for differences in protein or mitochondrial content between tissue types."
- We agree that single-volume lysate loading does not account for differences in protein or mitochondrial content between tissue types or samples. This is because this loading method for frozen tissue respirometry must be combined with tandem assessment of protein content and mitochondrial mass. To address the reviewer's comment we added a statement in the manuscript that the sample loading method proposed only works in tandem with normalization to total protein or mitochondrial content after running the samples.

- 2) Reviewer 2 states: "Reproducibility not shown: No inter-plate (repeatability) or intra-plate (CV) data to confirm method consistency. Would be interesting to see the assay performed on same samples in a different day. Considering the large number of samples available it would seem quite possible to plot inter-plate (different dates, same sample) Bland-Altman analysis seems possible. Similarly, it would be useful to see intra-plate coefficient of variation (%) from technical repeats of the same sample in the same plate."
- We appreciate the reviewer for bringing up the topic of reproducibility. We have interpreted this comment in two ways and have made accommodations for both interpretations. (1) If the intent of this comment was to ensure that we provide some reproducibility data from our validation study and to discuss methods to observe sample reproducibility in high-throughput applications, we have added a paragraph in the discussion and a new Supplemental Figure 3 to address these points. Specifically, we make mention to the fact that using aggregate measurements like the average of multiple control samples can provide a simple way to monitor reproducibility across plate and to make accommodations based on these trends. Supplemental Figure 3 shows this aggregate measurement along with %CV for the control samples ran in our validation study. (2) If the reviewer's concern is with the reproducibility of the RIFS (previous frozen tissue respirometry) methodology, this method has been cited and utilized by hundreds of groups on a wide variety of sample types and experimental setups since its publication and has been shown to be reproducible and reliable. The intent of this manuscript is to provide guidance on the utility of this respirometry protocol (and the traditional) in high-throughput experimental settings rather than an assessment of the RIFS protocol itself.
- 3) Reviewer 2 states: "Manual trace exclusion unclear: Criteria for removing traces are subjective and not fully described. I would suggest a mathematical method. For identifying outlier wells in Seahorse assays, a robust approach is to use the Median Absolute Deviation (MAD) across technical replicates and participants. Wells with M_i values exceeding a defined threshold (e.g., $M_i > 5$, Hoaglin 2013; Yopez et al. 2018) can be excluded. This provides an objective, reproducible method for outlier removal rather than relying on subjective manual trace exclusion."
- We thank the reviewer for bringing up the subject of our criteria for trace exclusion. While our data set and experimental setup necessitated a quality control to exclude samples at or near the limit of detection of the Seahorse instrument, we see great value in the reviewer's suggestion to utilize statistical tests for outlier identification in a data set like this. To address this comment, we have added more clear wording for how we defined our limit of detection and have cited manuscripts detailing our statistical rationales for assessment of limit of detection. Additionally, we have written a new paragraph in the discussion section to better inform the reader on the other types of statistical tests which could be used for automated quality control of their data set including Median Absolute Deviation as suggested by the reviewer.
- 4) Reviewer 2 states: "'High-throughput': While the authors report using multiple Seahorse XF96 instruments in parallel to reduce freeze-thaw cycles, it is unclear whether the same sample produces consistent results across different machines. Moreover, while this approach may be practical for their lab, most laboratories will only have access to a single Seahorse, limiting the general relevance of this 'high-throughput' claim. Including inter-instrument reproducibility data and clarifying the scalability limitations would strengthen the manuscript."
- We understand the reviewer's concern that most lab will not have access to more than one instrument for respirometry. To address the concern regarding instrument variability we have added a

new supplemental figure which shows that there was no significant difference between control samples ran on different instruments.

- 5) Reviewer 2 states: “No comparison to prior work: The method does not cite or acknowledge established approaches (e.g., Underwood et al., 2020).”
 - Per recommendation we have cited previously established approaches to higher throughput respirometry approaches for the utility of the reader.
- 6) Reviewer 2 states: “QC thresholds not validated: Acceptance criteria were empirically chosen from their dataset, without independent validation.”
 - We thank the reviewer for highlighting this point. While we understand that the reviewer would like a validation of our quality control thresholds, we feel that this fails to encompass the variability between tissues and experimental setups which may occur in respirometry experiments. As seen in Acin Perez et al. (2020) Figure 5J, the difference in activity for Complex I, II, and IV between different tissues is very large, therefore one set of quality control thresholds would not fit for all tissue types and experimental scenarios. Our manuscript guides the reader through our considerations and processes in developing a quality control protocol for our bespoke data set to serve as a guideline for making their own quality control, not as a pre-set to be applied to all respirometry data sets. To address the reviewer’s comment we have made sure to mention in the discussion that while we guide our readers to create their own quality control metrics, our specific metrics are bespoke to our study and do not represent a rule to be followed for all tissues and studies.
- 7) Reviewer 2 states: “Potential for bias: Fixed-volume loading could over- or under-represent OCR in different tissue types, affecting biological interpretation.”
 - We thank the reviewer for bringing up this point. While we understand the concern that a fixed-volume loading method could result in higher or lower OCR values, normalization to total protein or mitochondrial content will correct for potential bias. To address this, we have ensured that it is clear in the manuscript that fixed loading volume loading must be accompanied by normalization to total protein or mitochondrial content to be used for a study.

Reviewer 3:

Reviewer 3 states: “This article discusses novel methodologies, approaches and considerations to scale mitochondrial respirometry studies to hundreds to thousands of samples. Furthermore, the authors validated their high-throughput applications with 703 human brain samples.”

- We appreciate the kind comments from reviewer 3 regarding the important and timely nature of our manuscript. They provide no comments to be addressed in our revisions.

Thank you very much for your consideration and your help in improving this manuscript for inclusion in *Life Science Alliance*. We look forward to hearing back from you and your team.

Dr. Corey Osto, Dr. Orian Shirihai, and Dr. Linsey Stiles

February 19, 2026

RE: Life Science Alliance Manuscript #LSA-2025-03446RR

Dr. Corey A Osto
University of California, Los Angeles
Molecular and Medical Pharmacology
10833 Le Conte Avenue, Los Angeles, CA 90095
CHS 23-100
Los Angeles, CA 90095

Dear Dr. Osto,

Thank you for submitting your revised methods manuscript entitled "Novel Optimizations Allow for High-throughput Respirometry". We would be happy to publish your paper in Life Science Alliance pending final revisions necessary to meet our formatting guidelines.

MANUSCRIPT ORGANIZATION AND FORMATTING:

To avoid unnecessary delays in the acceptance and publication of your paper, please read the following information carefully. Full guidelines are available on our Instructions for Authors page, <https://www.life-science-alliance.org/authors>

- If possible please provide an institutional email address for the corresponding author.
- For brevity and to reduce redundancy (i.e. the term "novel") we suggest changing the title to "Optimizations Enabling High-throughput Respirometry" or similar.
- Please upload your Tables in an editable .doc or Excel format, and be sure to label them. They can be included at the bottom of the main manuscript file or sent as separate files.
- Please mark the corresponding author on the manuscript file
- Please add the X and Bluesky handles of your host institute/organization, as well as your own, and/or one of the authors, in our system.
- Please consult our manuscript preparation guidelines <https://www.life-science-alliance.org/manuscript-prep> and make sure your manuscript sections are in the correct order and that they are labeled correctly.
- It is recommended to exclude figures from the manuscript text and upload them separately.
- Please add your main, supplementary figure, and table legends to the main manuscript text after the references section.
- Please include a "Data Availability" section placed after the Materials & Methods section. Please consult our guidelines at <https://www.life-science-alliance.org/manuscript-prep#format>
- Please add an Author Contributions section to your main manuscript text.
- Please add a Conflict of Interest statement to your main manuscript text.
- There is a call-out in the manuscript text for figure 4D, and this figure doesn't have panel D. Please correct.

LSA encourages authors to provide a 30-60 second video where the study is briefly explained. We will use these videos on social media to promote the published paper and the presenting author (for examples, see <https://docs.google.com/document/d/1-UWCfbE4pGcDdcgzcmiuJI2XMBJnxKYeqRvLLrLSo8s/edit?usp=sharing>). Corresponding or first-authors are welcome to submit the video. Please submit only one video per manuscript. The video can be emailed to contact@life-science-alliance.org

FINAL FILES:

The following items are required for acceptance.

The license to publish form must be signed before your manuscript can be sent to production. A link to the license to publish form will be available to the corresponding author only. Please take a moment to check your funder requirements.

Thank you for your attention to these final processing requirements. Please revise and format the manuscript and upload materials as soon as you are able.

Thank you for this interesting contribution to the literature. We look forward to publishing your paper in Life Science Alliance.

Sincerely,

Reviewer #1 (Comments to the Authors (Required)):

My concerns have been fully addressed. I expect this work to be useful contribution. Congratulations.

Reviewer #2 (Comments to the Authors (Required)):

Thank you for the revisions. The authors have satisfactorily addressed my comments, and I have no further concerns.

February 27, 2026

RE: Life Science Alliance Manuscript #LSA-2025-03446RRR

Prof. Orian S. Shirihai
University of California, Los Angeles
Department of Molecular and Medical Pharmacology, David Geffen School of Medicine
650 Charles East Young Drive South
CHS 27-200
Los Angeles, CA 90095

Dear Dr. Shirihai,

Thank you for submitting your Methods article entitled "Scalable Workflows for High-Throughput Respirometry". It is a pleasure to let you know that your manuscript is now accepted for publication in Life Science Alliance. Congratulations on this interesting work.

Your article will publish open access upon publication under a CC-BY license.

DISTRIBUTION OF MATERIALS:

Again, congratulations on a very nice paper. I hope you found the review process to be constructive and are pleased with how the manuscript was handled editorially. We look forward to future exciting submissions from your lab.

Sincerely,
